# Pregnancy and Luteal Responses to Embryo Reinsertion following Embryo Flushing in Donor Mares

**DOI:** 10.3390/ani14111605

**Published:** 2024-05-29

**Authors:** Rebeca Martínez-Boví, Jana T. H. Gaber, Laura Sala-Ayala, María Plaza-Dávila, Juan Cuervo-Arango

**Affiliations:** Equine Fertility Group, Facultad de Veterinaria, Universidad Cardenal Herrera-CEU, CEU Universities, 46115 Alfara del Patriarca, Spain; rebeca.martinez@uchceu.es (R.M.-B.); jana.gaber2@alumnos.uchceu.es (J.T.H.G.); laura.sala1@alumnos.uchceu.es (L.S.-A.); maria.plaza@uchceu.es (M.P.-D.)

**Keywords:** horse, conceptus transfer, progesterone, endometritis, capsule fragment

## Abstract

**Simple Summary:**

**Simple Summary:** Embryo reinsertion into the same donor mare after embryo flushing would be useful for some reproductive techniques. However, this procedure has been unsuccessful in the past. This study envisages to gain insights into the mechanisms by which embryos fail after embryo flushing and reinsertion in the same mare. The results of this study showed that embryo mortality after embryo reinsertion is not due to luteal deficiency. In contrast, it seems that changes in the endometrial environment due to bacterial contamination and/or inflammation during the uterine lavage are more likely to be responsible for the destruction of most reinserted embryos.

**Abstract:**

The effect of embryo reinsertion immediately after embryo flushing was studied. In Experiment 1, eight mares were used during 32 cycles (8 cycles in each group). For the first two groups, inseminated mares were flushed 8 days after ovulation and prostaglandin F2α was not administered: in group EF-ET (embryo flushing and embryo transfer) the embryo was reinserted in the same donor mare, while in the EF group, no further procedure was performed. In the third group (ET), non-inseminated mares (recipients) received a Day 8 embryo. Progesterone concentration was measured before EF/ET and 72 h after in the three groups. In Experiment 2, twelve mares were used during 17 cycles in two groups, EF-ET (n = 11) and ET (n = 6), as in Experiment 1, except that every mare was flushed 24 h after embryo transfer to retrieve the embryo. Fewer pregnancies resulted after transfer in EF-ET cycles (0/8, 0%) than in the ET group (6/8, 75%). Progesterone concentration decreased significantly (*p* = 0.05) 72 h after EF-ET but not in EF or ET cycles (*p* > 0.1). Three mares from the EF-ET showed full luteolysis and signs of endometritis. In Experiment 2, more (5/6; *p* = 0.08) grade 1 embryos were recovered in the ET compared to the EF-ET group (3/7); 4 embryos were graded 3–4 (were broken or had signs of degeneration) in the EF-ET group but none in the ET group. In both groups, capsule fragments were obtained as indicative of the presence of a recently destroyed embryo in the EF-ET (n = 3) and ET (n = 1) groups. Positive bacterial cultures were obtained in 2/11 and 1/6 embryo flushes from the EF-ET and ET groups, respectively.

## 1. Introduction

Embryo flushing and immediate embryo reinsertion has been largely unsuccessful in mares [1,2]. This procedure would be useful to collect and examine embryos in vitro for preimplantation genetic diagnosis [3,4,5] before reinserting them in horse breeds in which embryo transfer is not allowed (i.e., Thoroughbred), or in donor mares with twin ovulation in which only one recipient is available, and the owner wishes to have the donor mare carrying her own pregnancy. Furthermore, it would be a valuable tool to investigate the reproductive mechanisms behind the establishment or failure of pregnancy to gain insights into the embryo–maternal communication and recognition of pregnancy [6], or to improve the efficiency of assisted reproductive techniques, such as embryo transfer [7,8].

Several studies [9,10,11] have shown the possible outcome of pregnancy establishment following unsuccessful embryo flushing (negative embryo recovery). The likelihood of unwanted pregnancy following a negative embryo flushing can be significant (around 20%) when PGF_2α_ is not administered following embryo flushing [12], which contrasts with the difficulty of reinstating a pregnancy after embryo flushing and reinsertion [1,2].

It is unknown whether reinserted embryos fail due to PGF_2α_ release and endometrial inflammation from the uterine lavage procedure or accidental bacterial contamination during embryo transfer. On the other hand, full luteolysis and a reduction in progesterone concentration to basal levels does not appear to be a cause of embryo failure following embryo reinsertion, since most mares maintain the CL and even show prolonged dioestrus following embryo flushing [1,12,13,14].

Larger embryos are more fragile than smaller embryos (i.e., Day 9 vs. Day 8), and therefore the size of the reinserted embryo could play a role in the likelihood of success [15]. In fact, the embryo reinsertions reported in a previous study were remarkably large (Day 10.5 to 13.5), of which only one (1/23) was carried out to term [2]. Furthermore, bacterial contamination during embryo reinsertion may be difficult to avoid since rectal manipulation of the uterus during uterine massage while the Foley catheter connects the vulva with the anterior vagina is likely to facilitate the passage of bacteria to the cervix, which in turn can be introduced along the embryo transfer gun [16].

The objectives of this study were to determine the effects of embryo reinsertion immediately after embryo flushing 7–9 days after ovulation on progesterone concentration, embryo growth and morphology, pregnancy outcome, and endometrial bacterial contamination in the mare.

## 2. Materials and Methods

### 2.1. Animals

Twelve non-lactating mares (Equus caballus) from different breeds (Arabians, Spanish purebred, and crossbred) weighing 420 to 670 kg and aged 5 to 20 years old (mean 11.2 ± 5.5 years old) were used in the study during two breeding seasons. Mares belonged to the research herd of the University Cardenal Herrera-CEU, located in Náquera, Spain (39°39′ N), and were kept in sand paddocks in groups of 3 to 4 animals and were fed on hay and cereal concentrate three times a day, with ad libitum access to water and mineralized salts. All mares were cycling at the beginning of the study. Mares chosen for the study had no ultrasonographic signs of ovarian or uterine abnormalities [17]. Furthermore, a Spanish purebred stallion aged 12 years old of proven fertility was used as a semen donor for breeding. Animal procedures were approved by the local animal welfare committee of the Universidad CEU Cardenal Herrera and authorised by the regional official authority (Generalitat Valenciana) for the use of animals in research, with licence reference number 2023-VSC-PEA-123.

### 2.2. Experimental Design

The study was carried out between October 2022 and April 2024 and was organized in two experiments.

Experiment 1: Eight mares were used during 24 cycles (3 oestrous cycles each mare). Each cycle for every mare was allocated to one of the three different experimental groups in a crossover design:

EF (embryo flushing; n = 8): Inseminated mares were flushed 8 days after ovulation, and no further uterine manipulation was performed. The cycle was included in this group regardless of the outcome of the embryo flush (either positive or negative embryo recoveries).

EF-ET (embryo flushing + embryo transfer; n = 8): Inseminated mares were flushed 8 days after ovulation. Only positive embryo flushes were used in this group, as the embryo was reinserted within 30 min after recovery in the same donor mare by transcervical transfer.

ET (embryo transfer only; n = 8): Non-inseminated mares received a Day 8 embryo (conceived by a different mare) 5 to 8 days after ovulation, with embryo–recipient synchrony of +3 to 0 days; where the embryo could be between 3 and 0 days older than the uterine age (measured from the day of ovulation of the recipient mare).

For each group, the inter-ovulatory interval (IOI), the progesterone concentration before the beginning of the procedure (embryo flushing and/or embryo transfer) and 72 h after, and the pregnancy outcome were compared. Mares entering anoestrus before completing the set of 3 cycles for the experimental groups were used again after the period of winter anoestrus, once they had ovulated spontaneously the first time at the beginning of the breeding season. Similarly, in mares with endometritis, a “wash out or rested” cycle was allowed after proper treatment, before completing the remaining experimental cycles.

Experiment 2: This experiment was designed to gain insights into the effect of embryo reinsertion on embryo morphology and endometrial bacterial contamination, as no pregnancy was obtained from the previous experiment in the group of EF-ET. Furthermore, a technique used to reduce possible contamination during retransfer was used to prepare the vulva and vestibule before the EF and ET, to minimize contamination. Twelve mares were used for 17 cycles divided into two experimental groups:

EF-ET-EF (n = 11): Inseminated mares with a positive embryo flushing recovered 7–9 days after ovulation had their own embryo reinserted. Then, 24 h after reinsertion, the mare was flushed again by uterine lavage to attempt embryo recovery. The presence of bacterial contamination in the uterus was assessed by culturing an aliquot of the flushing fluid 24 h after embryo reinsertion (during embryo retrieval).

ET (n = 6): Recipient mares received a Day 7–9 embryo 5 to 9 days after ovulation (embryo–recipient synchrony +3 to 0 days) by transcervical transfer, and 24 h after ET, the mare was flushed again to attempt embryo recovery. Similarly, the presence of bacterial contamination in the uterus was assessed by culturing an aliquot of the flushing fluid.

For both groups, embryo morphology and embryo size were compared between the time of transfer and recovery; similarly, the percentage of mares with a positive bacterial growth were compared.

### 2.3. Ultrasound Examinations and Breeding Management

Mares were examined by transrectal ultrasonography once daily when in oestrus. When the mare presented obvious endometrial oedema according to the scoring system previously reported [18,19], in brief, scores of 2 to 3 (score of 0 = no endometrial folding; score of 1 = slight degree of endometrial folding; score of 2 = moderate degree of endometrial folding; and score of 3 = maximum degree of endometrial folding), and first showed a follicle of 30 to 40 mm in diameter, according to previous breeding records of individual mares [20], the mare was inseminated artificially (AI) with 1 billion motile and freshly collected sperm. Ovulation was induced at the time of AI with 200 µg of buserelin (Suprefact 1 mg/mL, Sanofi-Aventis Deutchland GmbH, Frankfurt, Germany) administered subcutaneously. If the mare had not ovulated by 72 h, AI was repeated. Ovulation was diagnosed by daily ultrasonography after AI (Day 0 = day when ovulation was first diagnosed) and confirmed 24 h later by the formation of an echoic corpus luteum (CL) [21].

### 2.4. Embryo Flushing

Embryo flushing (EF) was performed 7 to 9 days after ovulation (considering Day 0 = the day of the first ovulation, in cycles with asynchronous double ovulations) for donor mares and 24 h after reinsertion or embryo transfer (ET) in recipient mares. For these procedures, the mare was restrained in a stock, the perineum and vulva were washed and scrubbed three times with neutral soap during Experiment 1 and with chlorhexidine soap 0.8% (Desinclor, Antiseptic soap 0.8% chlorhexidine, Imark laboratories, Spain) in Experiment 2, and then rinsed with tap water. After that, the area was dried with paper towels and the entrance to the vestibule was cleaned with cotton wool soaked with sterile distilled water. A 32 FR Foley catheter (Embryo flushing catheter 32 CH, Minitube Iberica, Tarragona, Spain) connected to a Y tube closed system (Set of tubes Y Luer, Minitube Iberica, Tarragona, Spain), with one way connected to a 1 L plastic bottle of ringer’s lactate (Ringer lactato 1 L, Braun Vetcare, Rubí, Spain) and the other way to an embryo filter (Miniflush embryo filter, Minitube Iberica, Tarragona, Spain), was used to perform the EF. The Foley catheter was passed through the cervix using a sterile glove and once in the uterus, the balloon was inflated with 40 mL of air and the catheter was pulled backwards slightly to seal the internal os of the cervix. Then, 1 L of RL was infused in the uterus for each of the two flushing attempts. Flushing fluid was recovered with the aid of uterine massage. Once the second flushing attempt had been completed, the embryo filter was taken to the laboratory and searched thoroughly, using a stereoscope (Zeiss stemi 508 doc, Zeiss Ibérica, Bétera, Spain). Recovered embryos were rinsed with holding media (Equihold, Minitube Ibérica, Tarragona, Spain) and measured with a graduated scale and evaluated. Embryo grade (1 to 4) was assigned according to the scoring system described previously [22,23]. In brief, grade 1 embryos were completely round and had no imperfections; grade 2 embryos had slight imperfections; grade 3 embryos had moderate level of imperfections; and grade 4 embryos had severe problems that were easily identified, like rupture of the zona pellucida or capsule. Embryos were loaded in a sterile 0.5 mL straw and fitted in an embryo transfer pipette and gun (ET sheath and ET syringe for Day 8 embryos, IMV technologies, France). Embryos were transferred within 30 min of recovery.

In Experiment 1, no prostaglandin F2 alpha was administered following the embryo flushing, so that the IOI and pregnancy status could be determined.

### 2.5. Embryo Transfer

The recipient and donor (for embryo reinsertion) were prepared similarly. The vulva, perineum, and vestibulum were cleansed as described previously as per embryo flushing. Mares were sedated with 4 mg of detomidine i.v. (Domidine 10 mg/mL Detomidine Hydrochloride, Eurovet Animal Health, Bladel, The Netherlands). The embryo was transferred transcervically by a modified Wilsher’s ET technique [24] using a 35 cm long vaginal speculum (duckbill type vaginal speculum for horses, Kruuse, Denmark) and cervical forceps (Equivet cervix forceps for E.T., Kruuse, Langeskov, Denmark). Following ET, the tip of the ET pipette was rinsed with holding media into a petri dish and searched to ensure that the embryo did not stick to the exit hole. As the embryo transfer was performed with the aid of a vaginal speculum, representative images of a cervix from one mare from the ET group (Figure 1A) and from two mares from the EF-ET group (Figure 1B,C) were taken to depict the cervical morphology and the level of cervical inflammation (Figure 1).

### 2.6. Pregnancy Diagnosis

Pregnancy diagnosis was performed 3 days after embryo transfer or reinsertion and again 3 days later (14 days after ovulation of the embryo donor mare) by transrectal ultrasonography [17]. The pregnancy was terminated in pregnant mares by administering 5 mg of dinoprost i.m. (Dinolytic, dinoprost trometamol 5 mg/mL, Zoetis, Madrid, Spain). Similarly, non-pregnant mares which had not returned to oestrus spontaneously by 19 days after ovulation were assumed to be in prolonged dioestrus [25] and 5 mg of dinoprost was administered i.m. to induce luteolysis.

### 2.7. Bacteriology

In Experiment 2, an aliquot of the first litre of flushing fluid (24 h after embryo transfer) was collected in a 15 mL sterile tube as it was flushed out the uterus through the embryo filter. A drop of the aliquot was plated in blood agar (BD Columbia agar, 5% sheep blood, Becton Dickinson GmbH, Heilderberg, Germany) and incubated for 24 h at 37 °C for bacteriological culture and identification. Bacterial growth and morphology and the presence of haemolysis around bacterial colonies were evaluated macroscopically 24 h after incubation, as described previously in our laboratory [12]. In brief, plates with a haemolytic halo around the colonies were suspected of *Streptococcus equi zooepidemicus*, while mucoid white colonies were suspected of *Escherichia coli*.

### 2.8. Blood Samples and Progesterone Assay

In Experiment 1, the sampling protocol consisted of two samples taken, the first one just before the embryo flushing/embryo transfer, and a second sample 72 h later. The blood samples were collected from the jugular vein into heparinized tubes and centrifuged (2000× *g* for 10 min). Plasma was decanted and stored at −20 °C until it was assayed. The progesterone plasma concentrations were assayed using a competitive solid-phase ELISA (DRG Instruments, GmbH, Marburg, Germany). It was determined without extraction from plasma in duplicates. The assay sensitivity was 0.02 ng/mL, and the intra-assay CV was 3.8%.

### 2.9. Statistical Analyses

Sequential data (progesterone concentration, IOI, and embryo growth rate) were tested for normality using the Shapiro–Wilk test. The effect of the experimental group on the mean progesterone concentration at 0 and 72 h was tested by an unpaired *t*-test, while the difference in progesterone concentration between 0 and 72 h was tested by a paired *t*-test within groups. The difference in mean IOI (Experiment 1) and embryo growth rate (Experiment 2) between the groups was tested by the Mann–Whitney non-parametric test.

Fisher’s exact test was used to analyse categorical variables (pregnancy outcome, percentage of mares with full luteolysis, and signs of endometritis).

All data were computed in statistical software (Systat13 version 1, Santa Clara, CA, USA). A probability of *p* ≤ 0.05 indicated that a difference was significant, whereas probabilities between *p* > 0.05 and *p* < 0.1 indicated that a difference approached significance.

## 3. Results

### 3.1. Experiment 1

The likelihood of pregnancy in donor mares which had their own embryo reinserted (0/8, 0%) was lower (*p* < 0.01) than in recipient mares which had not been flushed previously (6/8, 75%; Table 1). The mean embryo diameter and percentage of grade 1 embryos for each group was not different (*p* > 0.05): 670 ± 230 µm and 610 ± 290 µm, and 87.5 and 75% of grade 1 embryos, for the EF-ET and ET groups, respectively.

The progesterone concentration at Hour 0 (before the cervical manipulation: embryo flushing, embryo flushing + embryo transfer, or embryo transfer) was not different amongst groups (*p* > 0.05; Table 1). However, 72 h after the procedure, the progesterone concentration was lower (3.9 ± 2.3 ng/mL) in the EF-ET group than in the ET group (6.1 ± 1.3 ng/mL; *p* < 0.05) but was similar to the EF group (5.9 ± 1.2 ng/mL; *p* > 0.05, Table 1). There was a significant decrease in progesterone concentration (−2.1 ± 2.7 ng/mL) between 0 and 72 h after the gynaecological procedure only in the EF-ET group (*p* = 0.05; Table 1). Three mares (3/8, 37.5%) from the EF-ET group had full luteolysis (P4 concentration < 2 ng/mL and obvious endometrial oedema, score of 1 to 2) 72 h after EF and ET, while no mare from the EF and ET groups showed signs of full luteolysis by 72 h. Similarly, more mares (4/8, 50%) from the EF-ET group developed free intrauterine fluid 72 h after the procedure compared to mares from the ET (0%, *p* = 0.07) and the EF group (12.5%; *p* > 0.1). Three of these four mares from the EF-ET had purulent (echoic) free intrauterine fluid (depth > 25 mm).

The IOI was not different (*p* > 0.1) between the EF (22.5 days) and the EF-ET groups (21.7 days). In the EF-ET group, the IOI could only be calculated from six mares, since one mare entered anoestrus following the EF-ET, and in the other mare, luteolysis was induced after entering prolonged dioestrus. The IOI was not calculated in ET mares, as only two mares were not pregnant after embryo transfer, and therefore, luteolysis had to be induced in the remining six mares.

### 3.2. Experiment 2

The percentage of grade 1–2 embryos (6/6, 100% for group ET and 10/11, 91% for group EF-ET) and the mean diameter of embryos transferred in mares from the EF-ET and ET groups was not different, 767 ± 456 µm (n = 11) and 630 ± 579 µm (n = 6), respectively (*p* > 0.1; Table 2). The embryo recovery attempts performed 24 h after ET resulted in a similar (*p* > 0.1) embryo recovery, 63.6% (7/11) and 83.3% (5/6) of embryos for the EF-ET and ET groups, respectively (Table 2). The morphology and grade of embryos recovered 24 h after EF and embryo reinsertion in the EF-ET group varied greatly (Figure 2). Three embryos (42.9%) were scored as grade 1–2 (Figure 2, panel 5B) and had doubled their size within 24 h of transfer in the EF-ET group, compared to 100% of embryos (5/5; *p* = 0.08) in the ET group (Figure 3, panel 1B). In the EF-ET group, four embryos were grade 3–4 and were either broken (n = 2; Figure 2, panels 3B and 4B) or had signs of degeneration (Figure 2, panel 6B), with a similar or reduced diameter 24 h after reinsertion. In contrast, no embryo was graded 3–4 in the ET group (0/5, *p* = 0.08; Table 2). Twenty-four hours after three and one embryo transfers from the EF-ET and ET groups, respectively, only fragments of the embryo capsule (no traces of the trophoblast) were found in the flushing filter (Figure 2, panels 1B and 2B; Figure 3, panel 2B). A piece of capsule fragment (Figure 2, panel 4c) was found also along with a broken embryo (Figure 2, panel 4B), which may indicate the order of events taking place in the process of embryo destruction: from an intact embryo, rupture of the capsule and trophoblast, disappearance of the trophoblast, and the capsule fragments remaining longer. One embryo reinsertion resulted in a negative flushing attempt 24 h after EF-ET, in which no trace of the embryo (not even capsule fragment) was found.

The fold increase in embryo diameter between ET and 24 h after tended to be lower (*p* = 0.06) in reinserted embryos (1.6 ± 0.8-fold increase) than in mares of the ET group (2.1 ± 0.4-fold increase; Table 2).

The bacteriological culture of the recovered fluid 24 h after the first set of ET and embryo reinsertions resulted in positive growth by 48 h of culture in only two samples of the EF-ET mares (2/11) with pure growth of the streptococci spp. beta haemolytic; these two positive samples corresponded to the recoveries of one embryo with signs of degeneration and reduced growth 6B (Figure 2, panel 6B) and one apparently normal grade 1 embryo (Figure 2, panel 5B), while one sample of the ET mares (1/6) resulted in a mixed growth of 4 colonies of streptococci spp. beta haemolytic and 30 colonies of *Escherichia coli*, corresponding to one apparently normal grade 1 embryo (Figure 3, panel 1B).

## 4. Discussion

The results of this study confirmed the disappointing pregnancy results obtained in previous studies [1,2] following embryo reinsertion. However, on this occasion, the embryos were smaller (younger embryos, Day 7 to 9) than in the previous study [2] in which large Day 10.5 to Day 13 embryos were used. Therefore, the fragility of large embryos does not appear to account for the lack of embryo viability following reinsertion in the same donor mare in which the embryo was flushed out. This is the first study, however, to show the effect of embryo reinsertion on the morphology and quality grade of embryos recovered shortly after reinsertion (by 24 h). These results gave insights about the timing of embryo disappearance and the macroscopic process of embryo demise following transfer into a suboptimal endometrial environment induced by uterine lavage and embryo transfer.

The reduction in progesterone concentration below the required levels (i.e., >2 ng/mL) to maintain pregnancy [26] did not seem to be the reason for embryo mortality, because five out of eight mares from the EF-ET group had normal dioestrus levels of progesterone (4 to 10 ng/mL) and inter-ovulatory intervals (i.e., 20 to 22 days), equivalent to that from recipient mares receiving an embryo observed in the current and previous studies [27]. Cervical and uterine manipulation during embryo flushing per se do not appear to cause enough oxytocin and prostaglandin release to induce full luteolysis and return to oestrus [1,12,13] to explain embryo loss due to luteal insufficiency after embryo reinsertion. In fact, several unwanted pregnancies have been reported following a negative flushing in embryo donor mares in which the embryo was left in the uterus at the time of flushing [9,12]. However, whether subtle and transient inflammation caused during uterine lavage could affect embryo development and survival without inducing luteolysis is unknown. Even if the uterine lavage-induced endometrial inflammation was not enough to directly kill the embryo, it could affect the gene expression profile and therefore alter the histotroph and endometrial secretome necessary for adequate embryo nutrition and development [28,29]. Furthermore, whether subtle changes in progesterone concentrations following embryo reinsertion might have influenced the process of maternal recognition of pregnancy is unknown but cannot be ruled out.

On the other hand, bacterial contamination during embryo reinsertion and the subsequent development of endometritis could have had a negative impact on embryo survival. Although only three mares from Experiment 1 showed signs of clinical endometritis (full luteolysis and presence of purulent intra uterine fluid), subclinical bacterial endometritis could not be ruled out [30] as a cause of embryo death following reinsertion. In Experiment 2, the vulva and perineum were cleansed using an antiseptic soap, in contrast to Experiment 1 in which only neutral soap was used to prepare the perineal area. However, two bacterial cultures were positive (out of 11 embryo reinsertions) despite a more sterile preparation of the vulva before EF and embryo reinsertion. Whether these positive samples were truly bacterial endometritis or just bacterial contamination during the sampling procedure is difficult to know [30]. Furthermore, the other nine negative samples could have been false negatives, due to the dilution of bacteria into the large flushing volume. Nevertheless, three (3/11) apparently normal embryos were recovered 24 h after embryo reinsertion. These embryos were grade 1 and had grown at a similar rate (2- to 2.3-fold increase) as control embryos (transferred into synchronized recipient mares). Unfortunately, it is unknown whether these three embryos would have continued developing normally to establish a pregnancy, had they not been flushed out 24 h after reinsertion. In a previous study [2], the growth rate of 6/11 (54%) Day 10.5 to 13.5 reinserted embryos was similar to that of controls, with 5 of them dying and disappearing 2 to 4 days after embryo reinsertion and only one being carried out to term. Therefore, normal embryo growth rate is not always and indicator of a healthy developing pregnancy.

An interesting finding of this study was to observe how quickly embryos can disappear or be destroyed after embryo transfer. In fact, only capsule fragments (sign of a destroyed embryo) were found in 3 of 11 reinserted embryos. Furthermore, 24 h after the transfer of six embryos into an apparently optimal endometrial environment of a properly synchronized recipient mare, one embryo was already destroyed. This confirms the observation of a previous study in which a timing of 48 h after transfer was likely to be too long to recover embryos which did not resist a suboptimal environment [31].

Further research is needed to elucidate whether the negative effect of embryo reinsertion on embryo viability is due to changes in the histotroph and endometrial secretome, which allows for normal embryo nutrition and development, and/or to bacterial-induced endometritis originating from contamination during embryo reinsertion. Continued research would shed light on the paradoxical observation of why some donor mares remain pregnant after an embryo flushing with a negative embryo recovery, while embryo reinsertion hardly results in a viable pregnancy when transferred into the same donor mare.

## 5. Conclusions

In conclusion, the reinsertion of Day 8 embryos resulted in no viable pregnancy. This was not due to luteal deficiency. Reinsertion of Day 7–9 embryos into the same donor mares resulted in a variety of effects on embryo morphology and viability within 24 h of transfer, in which 3/11 embryos developed normally (similar to controls), but the remaining embryos were destroyed or showed signs of degeneration.

## Figures and Tables

**Figure 1 animals-14-01605-f001:**
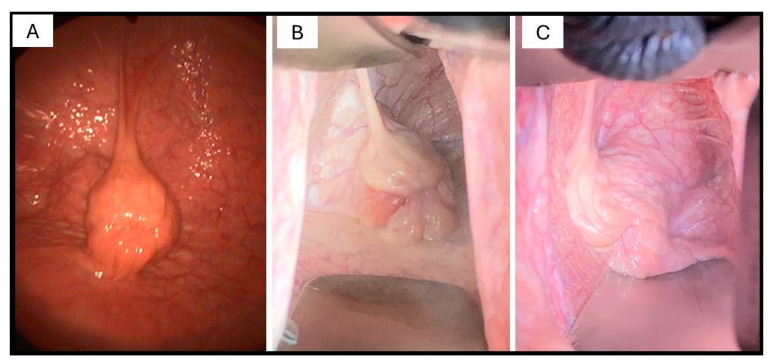
Representative images taken just before ET (panel **A**) from a recipient mare (ET group; not previously flushed) and before embryo reinsertion in two mares (panels **B**,**C**) which had been flushed previously. The diameter of the cervix and the degree of oedema of the cervical folds around the external os of the cervix appear greater in mares (**B**,**C**) compared to (**A**).

**Figure 2 animals-14-01605-f002:**
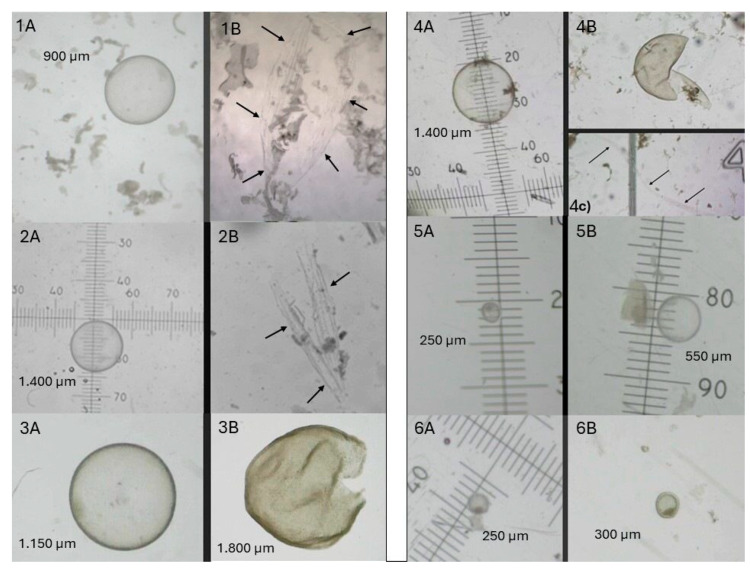
Light stereo-microscope representative images of different outcomes following embryo transfer of Day 7–9 embryos from EF-ET mares before (panels **1A**,**2A**,**3A**,**4A**,**5A**,**6A**) and 24 h after embryo reinsertion (panels **1B**,**2B**,**3B**,**4B**,**5B**,**6B**) in the same donor mare. Images were taken at different magnification; therefore, the diameter of each embryo is indicated. Black arrows depict the position of the capsule fragments recovered 24 h after embryo reinsertion. Embryos 1A and 2A resulted in the recovery of capsule fragments only 24 h after ET (panels **1B**,**2B**); Embryos 3A and 4A were ruptured 24 h after transfer (**3B**,**4B**), with a capsule fragment (**4C**) found along the embryo. Embryo 5A grew at the expected rate 24 h after reinsertion and maintained the grade 1 (**5B**). Embryo 6A hardly grew 24 h after reinsertion and showed signs of degeneration (**6B**).

**Figure 3 animals-14-01605-f003:**
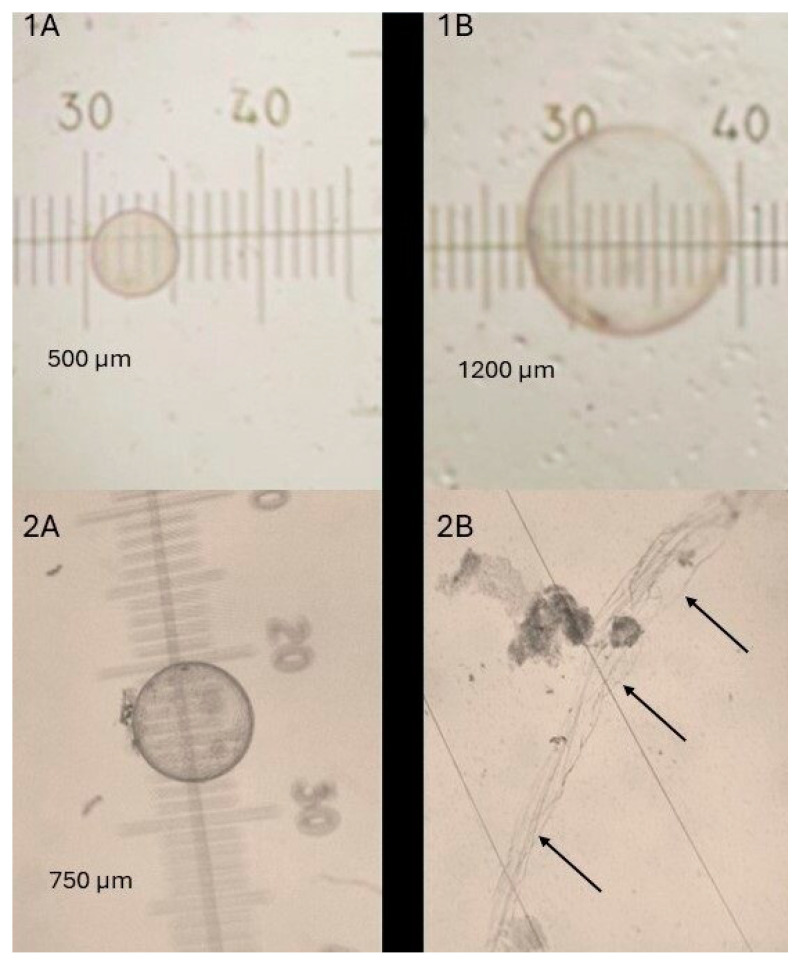
Light stereo-microscope images of the two outcomes following ET and embryo recovery in recipient mare receiving a Day 7–9 embryo from a different donor mare. The embryo diameter is shown in each image. A grade 1 (panel **1A**) embryo doubled its size by 24 h (panel **1B**) and maintained the same score (grade 1). A grade 2 embryo (with darker areas within the trophoblast, panel **2A**) disappeared 24 h after, in which only a capsule fragment could be retrieved (the recovered fragment is depicted by black arrows, panel **2B**).

**Table 1 animals-14-01605-t001:** Reproductive characteristics of mares from Experiment 1.

Group	n	P4 0 h (ng/mL)	P4 72 h (ng/mL)	P4 Difference (ng/mL)	Free IUF by 72 h (%)	Pregnancy after ET (%)	IOI (Min–Max)
EF	8	6.7 ± 2.3	5.9 ± 1.2 ^a,b^	−0.8 ± 1.3	12.5	-	22.5 ± 1.3 (20–24)
EF-ET	8	6.0 ± 1.2	3.9 ± 2.3 ^a^	−2.1 ± 2.7 *	50.0	0.0 ^a^	21.7 ± 4.1 (17–29)
ET	8	6.5 ± 1.4	6.1 ± 1.3 ^b^	−0.4 ± 1.5	0.0	75.0 ^b^	-

EF: Embryo flushing only; EF-ET: embryo flushing and embryo reinsertion by embryo transfer in the same donor mare; IUF: intrauterine fluid; ET: embryo transfer in recipient mares with an embryo from a different mare; P4: progesterone concentration; IOI: inter-ovulatory interval (days). Within column, different letters (^a,b^) indicate a significant difference (*p* ≤ 0.05). An asterisk (*) indicates a significant (*p* = 0.05) decrease in progesterone concentration between the moment before EF-ET and 72 h after.

**Table 2 animals-14-01605-t002:** Embryo characteristics following 24 h of transfer in mares from Experiment 2.

Group	n	Embryo Size at ET	Positive Recoveries	Fold Increase in Size by 24 h	Grade 1–2 Embryos	Grade 3–4 Embryos	Recovery of Capsule Fragments
EF-ET	11	767 ± 456	7/11 63.6	1.6 ± 0.8 ^a^	3/7 42.9 ^a^	4/7 57.1 ^a^	3/11 27.3
ET	6	630 ± 579	5/6 83.3	2.1 ± 0.4 ^b^	5/5 100.0 ^b^	0/5 0.0 ^b^	1/6 16.7

EF-ET: embryos were reinserted in the same donor mare immediately after embryo flushing (EF); ET: embryo transfer in recipient mares with an embryo obtained from a different donor mare. Day 7–9 embryos were recovered 24 h after ET. Within column, different superscripts (^a,b^) indicate a tendency approaching significance (*p* < 0.1).

## Data Availability

The original contributions presented in the study are included in the article, further inquiries can be directed to the corresponding author.

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
