# Peer review of "Pregnancy and Luteal Responses to Embryo Reinsertion following Embryo Flushing in Donor Mares"

_animals, 2024, doi:10.3390/ani14111605_

Round 1

Reviewer 1 Report

Comments and Suggestions for Authors

Page 1 line 8. Diagnosis is spelled incorrectly

Page 1 line 28. The “(“ should be a “/”

Page 1 lines 39-44 is the same as simple summary and is also a bit too long.

Page 2 line 82 insert “a” between as and semen.

Page 2 line 91. This says mares were allocated to one of four groups but only 3 are described.

Page 2 line 99.  What is meant by “shortly?”  5 min, 1 hour, 5 hours?  This should be clearly spelled out even if it is a range.

Page 3 line 102.  This line stated the embryo could be between 3 and 0 days younger than the uterine age.  Would it not be the opposite?  If the mare is day 5-8 from ovulation but the embryo is consistently 8 days old, won’t the embryo be 3-0 days older than the uterine age?

Page 3 Line 127-128. Please just include a brief description of the scoring system or at least which score was considered the appropriate level of edema. It is so annoying for readers to have to track down more papers to figure out what was done.

Page 3 Line 128-129.  Can you clarify this?  You used the previous records for each particular mare to determine if a 30 vs a 40mm follicle was likely to respond to buserelin? What is the citation for, using this as a method for determining a preov follicle?

Page 3 line 130 “foar” should be “for”

Page 4 line 156 “rinse” should be “rinsed”

Page 4 lines 157-158, again, it would be appreciated if the scoring system was just briefly described so readers didn’t have to go find another paper. 

Page 4 Line 184 Was it always 14 days after ovulation?  It seemed as if the recipients varied by day they received an embryo.

Page 5 Lines 197.  Again, just describe it.

Page 5 line 200. Could this read “The sampling protocol consisted of two samples, the first one just before embryo flushing/embryo transfer, and the second sample 72 hours later.” I would also move the centrifugation information after this sentence so that the series of events stays together.

Results: In experiment 2, embryo grades are referenced as both indo-arabic and roman numerals.  It would be ideal to choose one and use it consistently.

Page 9 Line 334 histotroph is misspelled.

Page 9 line 343 Change “still came positive” to “ were positive”

Page 9 Lines 350-352.  I think there is a word missing in this sentence.  Putting a “they” between “had” and “not”

Page 9 355-359.  This sentence is a little confusing.  It starts with priming of estrogens and ends with mare being treated with progesterone.  A mare could have a uterus that was primed with estrogens and still need exogenous progesterone if she has luteal insufficiency for some reason. Related to that, why is it assumed that a mare needing progesterone has a suboptimal environment, I think that would depend on how quickly she was started on exogenous progesterone.  Again, this is where expanding on a citation rather than just dropping it in would be helpful for readers. 

Page 9 Lines 360-361.  Is it possible they are being destroyed during the embryo transfer, especially with some of the larger embryos?

Page 9 Lines 361-364. This sentence is a little long and confusing.  I think moving the by only 24h needs to be  moved up or this needs to be combined into 2 sentences.

Comments on the Quality of English Language

The English is very good with only minor editing required.

Reviewer 2 Report

Comments and Suggestions for Authors

Pregnancy and Luteal Responses to Embryo Reinsertion Fol- 1 lowing Embryo Flushing in Donor Mares

Keywords: horse; embryo reinsertion; progesterone; endometritis; capsule fragment

There are key words in the title. These should only appear in one place.

The abstract has 321 words. Must have a maximum of 200.

In line 139? Is correct??”... Foar these 139 procedures, the…”

The conclusion is more for discussion. This must be objective and robust. And you must not show or discuss results. This is discussion.

Reviewer 3 Report

Comments and Suggestions for Authors

In this manuscript, the authors explore the use of mare embryo donors as recipients of their own embryos, as previous works have shown that this technique was unsuccessful. At first the authors evaluate the effect of embryo reinsertion in pregnancy rates at Day 7-9, when compared to other recipients. Because results of experiment 1 confirmed poor results when the donor is used as a recipient, they evaluated in a second experiment the effect embryo reinsertion on luteal function, pregnancy outcome, endometrial bacterial contamination, and embryo morphology, in an attempt to further explain the poor pregnancy rates obtained. Although they did not reach a strong conclusion, the data obtained can be useful for further studies.

I have a few corrections:

“Experiment 1: eight mares were used during 24 cycles (3 oestrous cycles each mare). Each cycle for every mare was allocated to one of the four different experimental groups in a crossover design” – there are only three groups, please correct. Also, were the mares used in all groups? How did they go through all the groups? What about the mare that went in anestrous? And the ones that presented endometritis, they were treated and further used?

Line 109 – “Furthermore, a more “sterile” technique” – furthermore, a technique used to reduce contamination

Line 255 – “mares from EF-ET and ET group was not different, 767±456 µm (n=11) and 630±579 µm (n=6), respectively (p>0.1; Table 2)” – please state grade I embryo rates, as you did in experiment 1

Line 303 – when stating the figures in the text, numeration is wrong – Fig 1 is figure 2 and Fig 2 is figure 3, please correct

In the second paragraph of the discussion, the authors talk about P4 concentrations.  However, although the authors found P4 concentrations compatible with a normal diestrous, they found significant decrease in P4 concentrations 72h after the procedure, and this decrease could potentially influence embryo development and survival and indirectly affect maternal recognition of pregnancy, I think they could add this to the discussion.

Also, the authors state that the embryos were transferred within 30 minutes after recovery. Was it possible that some of the donors could still have some intrauterine fluid from the uterine flushing?  As some of the fluid is reabsorbed and some fluid is expelled, could some of the embryos be expelled with this fluid? This is just speculations and the authors are not required to add this to discussion.

Comments on the Quality of English Language

The manuscript is well written and does not require extensive edditing 
